# A pyrolysis-free Ni/Fe bimetallic electro-catalyst for overall water splitting

Ying Zang [1,2], Di-Qiu Lu[1], Kun Wang[2,3], Bo Li[3], Peng Peng [2] ✉, Ya-Qian Lan[1] & Shuang-Quan Zang [2] ✉

Catalysts capable of electrochemical overall water splitting in acidic, neutral, and alkaline solution are important materials. This work develops bifunctional catalysts with single atom active sites through a pyrolysis-free route. Starting with a conjugated framework containing Fe sites, the addition of Ni atoms is used to weaken the adsorption of electrochemically generated intermediates, thus leading to more optimized energy level sand enhanced catalytic performance. The pyrolysis-free synthesis also ensured the formation of well-defined active sites within the framework structure, providing ideal platforms to understand the catalytic processes. The as-prepared catalyst exhibits efficient catalytic capability for electrochemical water splitting in both acidic and alkaline electrolytes. At a current density of 10 mA cm$^{-2}$, the overpotential for hydrogen evolution and oxygen evolution is 23/201 mV and 42/194 mV in 0.5 M $H_2SO_4$ and 1 M KOH, respectively. Our work not only develops a route towards efficient catalysts applicable across a wide range of pH values, it also provides a successful showcase of a model catalyst for in-depth mechanistic insight into electrochemical water splitting.

Hydrogen energy, accepted as the ideal substitute for fossil fuels, plays an important role in energy transformation[1-4]. The large-scale application of hydrogen energy could reduce carbon emissions, generating revolutionary impact on all aspects of energy utilization[5-8]. Nowadays, electrochemical water splitting (EWS) is considered as the most green and sustainable method for hydrogen production[9,10]. Moreover, EWS could also facilitate the efficient consumption of renewable energy and the energy redistribution as the buffer to enhance the resilience of the energy system. Thus, designing and synthesizing of efficient electrocatalysts for water splitting have attracted numerous attention all over the world.

Generally, EWS involves two half-reactions, hydrogen evolution reaction (HER) and oxygen evolution reaction (OER)[10,11], which need efficient electrocatalysts to reduce the overpotential and accelerate the reaction. Traditionally, platinum and the oxides of iridium and ruthenium are the best candidates for catalyzing hydrogen and

oxygen evolution reactions, respectively[12-16]. Nevertheless, the high cost and scarcity strongly restrict their large-scale application. Meanwhile, single-function catalysts for HER or OER would inevitably increase the complexity of the water electrolysis equipment. Hence, dual-function catalysts with high activity for both HER and OER have opened an avenue for water electrolysis[17,18]. However, comparing to the highly efficient HER process with higher proton concentration, the OER process on anode usually suffers from the sluggish kinetics and rapid deactivation of catalysts in acidic conditions. Thus, most of the reported catalysts for EWS were performed in alkaline conditions and relatively high over-potential is needed to drive HER and OER reactions simultaneously in practical application[19]. Therefore, the research and development of dual-function catalysts with high activity for both HER and OER at various pH values is the key to achieving efficient hydrogen production from water electrolysis.

[1]School of Chemistry, South China Normal University, Guangzhou 510006, China. [2]Henan Key Laboratory of Crystalline Molecular Functional Materials, Henan International Joint Laboratory of Tumor Theranostical Cluster Materials, Green Catalysis Center, and College of Chemistry, Zhengzhou University, Zhengzhou 450001, China. [3]College of Chemistry and Pharmaceutical Engineering, Nanyang Normal University, Nanyang 473061, China. ✉e-mail: 13020010407@163.com; zangsqzg@zzu.edu.cn

By far, great efforts have been devoted to developing electro-catalysts with relatively low over-potential for water splitting. Catalysts such as LDH (layered double hydroxide)[20,21] and metal oxides/nitrides[22,23] have witnessed significant development for highly efficient EWS. Furthermore, considering that the partially embedded nano-particle active sites would result in a reduction of the catalytically active surface, various routes had been developed to improve the atomic utilization over the past decades. For example, derivatives of three-dimensional MOFs (Metal-Organic Frameworks) with open nanos-tructures, such as the nanosheets with rough surface and expanding layers[17], three-dimensional open nano-netcage electrocatalysts[24] and nanotube structures with well-defined inner channels and a large sur-face area[25] can effectively improve atomic utilization rate. Notably, catalysts with single-atomic active sites have also emerged and demonstrated promising potentials for overall water splitting[21,26]. To promote efficient hydrogen production via water splitting, it is vital to reveal the relationship between the structure and properties of cata-lysts, and provide instructions for material design. With rational design, controllable synthesis, and predictable structures, the compositions of the catalysts can reach an appropriate balance, which not only greatly improves the utilization efficiency of active sites, but also improves the activity of catalyst to a greater extent.

Pyrolysis indeed has been very widely used in the past decades. Unfortunately, pyrolysis process are always carried out in specific atmosphere with high temperature, which are very energy-intensive. Meanwhile, the isolated metal atoms on the precursors are very unstable and tend to agglomerate during the pyrolysis process, not only reducing the efficient utilization of metal active sites but also making it difficult to realize the precise control. Thus, the development of non-carbonized synthesis strategies for the preparation of efficient electro-catalysts is challenging and vital. It could be expected that the pyrolysis-free route would lead to precisely synthetic control and a better-maintained structure, holding the potentials to elucidate structure-function relationships for the guidance of optimum electro-catalysts.

Herein, given to our previous exploration[27,28], we developed a pyrolysis-free route to synthesize highly efficient electrocatalyst for both HER and OER. Specially, through ion exchange under microwave condition, we prepared a two-dimensional conjugated phthalocyanine framework (CPF) simultaneously contained single atomic Ni/N/C and Fe/N/C (termed as CPF-Fe/Ni). The mild synthetic route not only ensured the effectively reduce the formation of agglomeration, but also maintained the integrity of the structure. During catalysis, these bimetallic sites between adjacent layers could synergistically form a bridging catalytic center, which facilitate high bi-functional catalytic performance for overall water splitting in both acidic and alkaline electrolytes. Meanwhile, the conjugated frameworks provided a con-ductive skeleton for superior mass transfer and ensured the long-term stability. The pyrolysis-free preparation method provides an idea for the research and development of efficient and stable catalysts for EWS. With simple synthesis process and much lower preparation cost than that of noble metal catalyst, pyrolysis-free synthesis holds promising application prospects in the field of efficient hydrogen production. Moreover, comparing with CPF-Fe/Ni, catalysts with single-type active center demonstrated very limited catalytic property in either acidic or alkaline media. The huge promotion of CPF-Fe/Ni derived from the synergistic effects of dual active sites. Fortunately, the pyrolysis-free route ensured the well-defined structure and precise location of active centers, offering a good opportunity to elucidate the composition and formation of active sites.

## Results

### Synthesis and preparation

CPF-Fe and CPF-Ni with single metallic sites were synthesized through solvothermal methods according to previous reports. Typically,

1,2,4,5-tetracyanobenzene (TCNB) was used as the monomer and 1,8-diazabicyclo(5,4,0)undec-7-ene (DBU) was introduced as the catalyst. During the synthesis, ferric chloride or nickel chloride was added as the structure direction agent to facilitate the construction of the conjugated quasi-phthalocyanine framework. During the preparation of CPF-Fe/Ni with bimetallic sites, the different electronegativity of metal ions could result in the uneven dispersion of metallic sites if simultaneously involved in the construction of CPF. Hence, $Ni^{2+}$ was directly added into the synthetic system after the construction of CPF-Fe and continuously triggered the ion exchange reaction under a microwave reaction to generate the bimetallic conjugation of CPF-Fe/Ni (Fig. 1a). Detailed synthetic process could be found in the supplementary materials.

### Structural characterization

The as-obtained CPFs contained the characteristics belongs to phtha-locyanine. The wavenumbers around 1547, 1697, 1711, and 1757 $cm^{-1}$ in the Fourier transform infrared spectrum indicated the formation of the macrocyclic structure of phthalocyanine[29], which could be observed in all of the CPFs (Supplementary Fig. 1). The $^{13}C$ solid-state NMR further confirmed the carbon structure of the as-designed CPFs (Fig. 1b)[30,31]. The local chemical environment of each CPFs was studied through X-ray photoelectron spectroscopy (XPS). The XPS survey spectra (Supplementary Figs. 2 and 3) revealed the existence of C, N, and the metal elements which were just as expected, while the high-resolution spectra indicated the homogenous state of Fe. Moreover, Fe atoms in CPF-Fe/Ni and raw CPF-Fe demonstrated same peaks in the high-resolution XPS spectra (Fig. 1c), suggesting that the ion exchange with small amount of Ni atoms didn't affect the coordination state of Fe. The N-coordinated metallic centers and the conjugated connec-tions facilitated the formation of active sites with effective charge transfer capability. Meanwhile, CPF-Fe/Ni exhibited a two-dimensional slice morphology under scanning electron microscopy (SEM) and transmission electron microscopy (TEM) (Fig. 1d, e). Besides, none of the aggregated metal particles was observed along the surface of CPF-Fe/Ni (Fig. 1e, f). Furthermore, we performed high-angle annular dark field scanning transmission electron microscopy (HAADF STEM) to reveal the atomic structure of CPF-Fe/Ni. In the aberration corrected images it could be found that the coordinated Fe and Ni sites were exclusively dispersed in the single-atom format and uniformly anchored throughout the surface (Fig. 1f). The associated elemental mapping also confirmed the uniform distribution of Fe and Ni atoms (Fig. 1g–i).

The coordination status and bonding configurations of Fe and Ni in CPF-Fe/Ni were deeply investigated by the synchrotron-based XANES and the extended X-ray absorption fine structure (EXAFS) spectra. According to previous reports, the pre-edge peak could be attributed to a 1s-4pz shakedown transition characteristic for a square-planar configuration with high D4h symmetry[32,33]. In this work, it should be noted that the pre-edge profile of CPF-Fe/Ni is similar to that of iron phthalocyanine (FePc) and nickel phthalocyanine (NiPc) (Fig. 2a, c), but exhibits a slight difference in intensities. These phe-nomenon suggested that the Fe/Ni center is coordinated with four N atoms by achieving a square-planar Fe/Ni-N4 molecular structure[34–36]. The Fe K-edge EXAFS spectra of CPF-Fe/Ni (Fig. 2b) displayed a Fe peak centered at 1.48 Å, which closed to that of FePc (1.50 Å), implying the presence of N-coordinated single Fe atom sites. Meanwhile, the peak at 2.2 Å that belonging to the Fe-Fe metallic bonds was missing in CPF-Fe/Ni, further confirming the atomically dispersed status of Fe. Similarly, with Ni, NiO, and nickel phthalocyanine (NiPc) as control samples, it could conclude that Ni was also existed as single atomic state in CPF-Fe/Ni similar to NiPc. Signals belonging to Ni-Ni metallic bonds were also absent in CPF-Fe/Ni. Hence, the coordinated Fe and Ni sites in CPF-Fe/Ni were well elucidated, offering an ideal model catalyst with superior capability (*vide infra*).

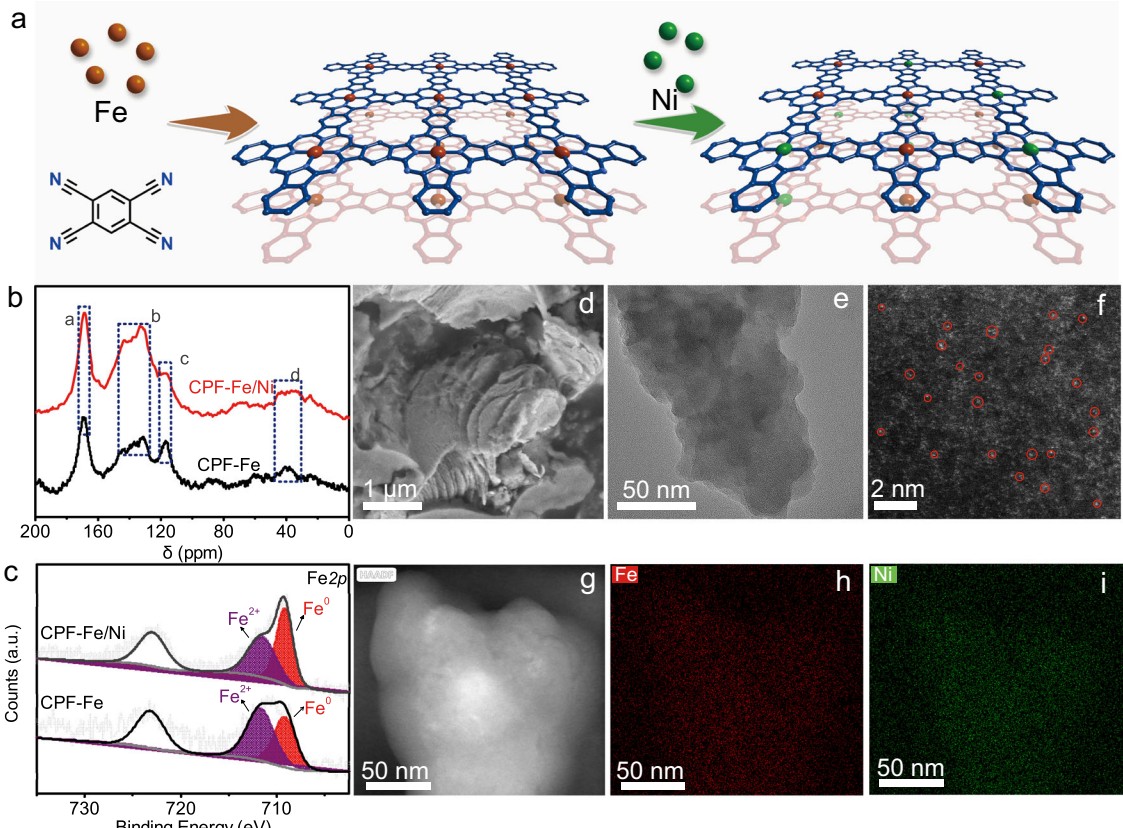

**Fig. 1 | Schematic illustration and structural characterization of CPF-Fe/Ni.**
**a** The synthesis process of CPF-Fe/Ni, **b** Solid-state CP-MAS $^{13}$C-NMR spectra of CPF-Fe and CPF-Fe/Ni, **c** High-resolution Fe2p spectra of CPF-Fe and CPF-Fe/Ni, the black curves are the fits data while the gray lines are the experimental results, **d** SEM and **e** TEM image of CPF-Fe/Ni, **f–i** HAADF STEM image and EDS-mapping results of CPF-Fe/Ni.

## Overall electrochemical water splitting

With the introduction of Ni, CPF-Fe/Ni exhibited excellent activity for both HER and OER in wide pH range. In the typical three-electrodes system, we performed the linear sweep voltammetry curves (LSV) of CPF-Fe/Ni and control samples (the blank carbon cloth, CPF-Fe, CPF-Ni, RuO$_2$, and 20% Pt/C) to evaluate the electrocatalytic activity for overall water splitting. Notably, CPF-Fe/Ni with bimetallic sites presented superior HER activity to that pristine CPF-Fe and CPF-Ni both in 0.5 M H$_2$SO$_4$ and 1 M KOH (Fig. 3a, b). CPF-Fe/Ni achieved a good HER activity with an onset potential near 0 mV. At the current density of 10 mA cm$^{-2}$, the overpotential in 0.5 M H$_2$SO$_4$ and 1 M KOH was only 23 mV and 42 mV, respectively, which were much lower than that of raw CPF-Fe (743 mV in 0.5 M H$_2$SO$_4$ and 202 mV in M KOH) and CPF-Ni (165 mV in 0.5 M H$_2$SO$_4$ and 161 mV in M KOH) (Fig. 3a). In addition, the corresponding Tafel plots calculated from the polarization data also suggested that CPF-Fe/Ni contained promoted reaction kinetics with a relatively smaller Tafel slop of 82.6 mV dec$^{-1}$ and 94.1 mV dec$^{-1}$ in 0.5 M H$_2$SO$_4$ and 1 M KOH, respectively (Supplementary Figs. 4 and 5). It could be found that the pure Fe sites held very poor catalytic activity for HER, while the activity of raw Ni sites was also limited. Hence, the joint of Ni and Fe sites generated electronic structures, leading to excellent HER catalytic activity of CPF-Fe/Ni.

The OER performance was also investigated in the same acidic and alkaline solution of 0.5 M H$_2$SO$_4$ and 1 M KOH (Fig. 3b, e). Similarly, CPF-Fe/Ni demonstrated excellent catalytic activity. Its overpotential at 10 mA cm$^{-2}$ is 201 mV and 194 mV in 0.5 M H$_2$SO$_4$ and 1 M KOH, respectively, which were much lower than that of the commercial 20% Pt/C and RuO$_2$. Furthermore, the Tafel slopes (Supplementary Figs. 6 and 7) of CPF-Fe/Ni were also pretty lower than raw CPF-Fe and CPF-Ni, suggesting better kinetics during OER process. Thus, it could be

concluded that the presence of bimetallic sites would efficiently adjust the catalytic centers of CPF-Fe/Ni and promote the synergistic effect, leading to much enhanced catalytic performance during overall water splitting. Even the reaction kinetics in the neutral electrolyte (1 M KCl) are slow due to the low concentration of adsorbed reactants, however, as HER catalysts the CPF-Fe/Ni demonstrates very small over-potential of 23, 76, 42, 147 mV in 0.5 M H$_2$SO$_4$, 0.05 M H$_2$SO$_4$, 0.01 M KOH, and 1 M KOH (Supplementary Fig. 8), respectively. Meanwhile, as OER catalysts the over-potential of 201, 345, 310, 194 mV was obtained in 0.5 M H$_2$SO$_4$, 0.05 M H$_2$SO$_4$, 0.01 M KOH, and 1 M KOH (Supplementary Fig. 9), respectively. Thus, these findings suggested the high efficiency of CPF-Fe/Ni as catalysts for the overall water splitting in wide-pH range.

The turnover frequency (TOF) of CPF-Fe/Ni was calculated based on the reported protocols (at an overpotential of 200 mV and 400 mV for HER and OER process, respectively) to reveal the intrinsic electrocatalytic activity[37]. Impressively, in 0.5 M H$_2$SO$_4$ CPF-Fe/Ni yields a TOF value of 3.16 s$^{-1}$ for HER and 1.12 s$^{-1}$ for OER, while in 1 M KOH the TOF 2.64 s$^{-1}$ for HER and 1.4 s$^{-1}$ for OER, which was superior to most of the reported catalysts (Supplementary Table 1).

Then, a three-electrode cell configuration was employed with Ag/AgCl as the reference electrode. Both the working electrode and the counter electrode are drop-coated with carbon cloth of CPF-Fe/Ni. During the catalysis reaction, the cathode and anode produced hydrogen evolution reaction and oxygen evolution reaction, respectively. The current of the cathode was collected to evaluate the catalytic activity and stability of CPF-Fe/Ni both as hydrogen evolution and oxygen evolution catalysts. As expected, the activity of CPF-Fe/Ni was quite stable during HER and OER process. At 10 mA cm$^{-2}$, the real-time currents could keep nearly constant over a continuous operation of

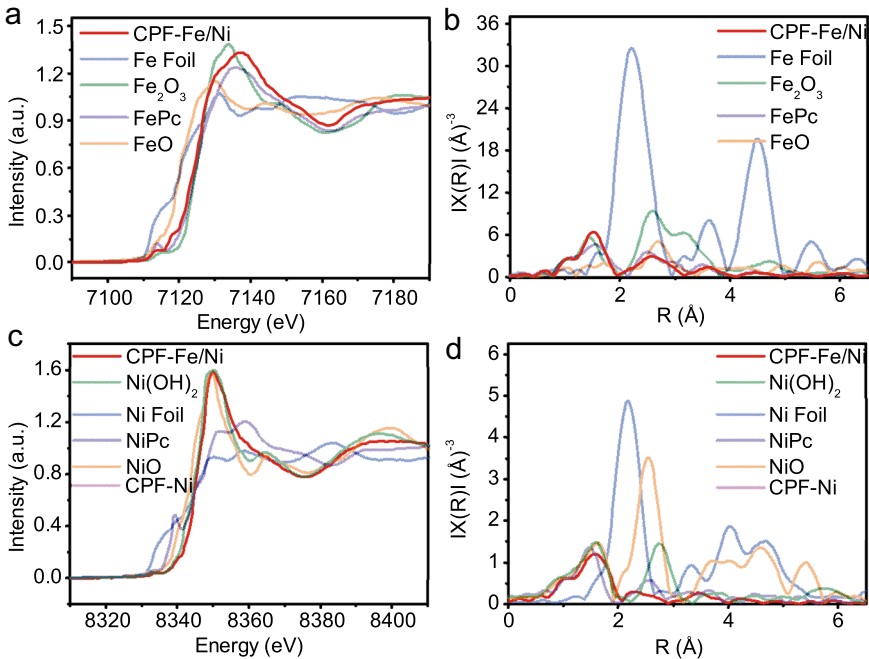

**Fig. 2 | The synchrotron-based XANES and the extended X-ray absorption fine structure (EXAFS) spectra. a** Normalized XANES and **b** radial structure functions (RSFs) of the Fe K-edge obtained by Fourier transformation $k^3$-weighted EXAFS results, with the Fe foil, $Fe_2O_3$, $Fe_3O_4$, and FePc serving as references; **c** Normalized XANES and **d** radial structure functions (RSFs) of the Ni K-edge obtained by Fourier transformation $k^3$-weighted EXAFS results, with the Ni foil, $Ni(OH)_2$, NiO, and NiPc serving as references.

200 h (Fig. 3c, f) in both acidic and alkaline solution. It was noteworthy that CPF-Fe/Ni exhibited a remarkable stability under a high current density of 600 mA cm$^{-2}$ in both acidic and alkaline solution (Supplementary Figs. 10 and 11). After electrochemical testing, the slice morphology (Supplementary Figs. 12 and 15) and the single-atom format (Supplementary Figs. 13 and 16) of CPF-Fe/Ni were all well maintained no matter under acidic or alkaline condition. Meanwhile, the FT-IR (Supplementary Figs. 14 and 17) and ICP (Supplementary Table 2) results also confirmed that the configuration and the structure of CPF-Fe/Ni were well reserved, confirming there is no any degradation of the catalysts active sites and the considerable stability in the electrochemical testing process. High-resolution N1s and Fe2p spectra of CPF-Fe/Ni after electrocatalysis testing in 0.5 M $H_2SO_4$ and 1 M KOH were collected (Supplementary Figs. 18 and 19), which also showed a highly consistent with other measurement and further confirmed the stability of elemental state.

Furthermore, we assembled a two-electrode cell with CPF-Fe/Ni for the practical utilization of water electrolysis in wide pH range. The catalysts were loaded on the carbon cloth and directly used as both anode and cathode. Noteworthy, this home-made cell coasted a potential of only 1.44 V and 1.57 V to reach 10 mA cm$^{-2}$ in 0.5 M $H_2SO_4$ and 1 M KOH solutions, respectively (Supplementary Figs. 20 and 21), which was very easy to be driven. The operation voltage for the CPF-Fe/Ni||CPF-Fe/Ni cell was much lower than the reported electrocatalyst based assembled symmetrical cells (Supplementary Table 1) in both acid and alkaline electrolyte. The faradic efficiency (FE) was further calculated by the water-drainage experiment[38], which was performed in a sealed H-type cell wherein CPF-Fe/Ni was acted as both cathodic and anodic electrodes (Supplementary Figs. 23 and 25, Videos 1 and 2). The working current density was set at 30 mA cm$^{-2}$ for 11 and 25 min in acid and alkaline electrolyte, respectively. By quantitatively collecting the generated $H_2$-$O_2$ gases upon water splitting and plotting the gas volume by time, the practical volume ratio of the collected $H_2$ to $O_2$ gas was obtained close to 2:1, which was consistent with the theoretical value (2:1). Based on these experimental and theoretical values, the FE

was then estimated to be 100% (Supplementary Figs. 22 and 24), which indicated that no side reaction took place during the overall water splitting electrolysis. Moreover, the cell established impressive stability. At 10 mA cm$^{-2}$, the current density of the cell could maintain long-term stability over 120 h without voltage decay in a wide pH range (Fig. 4b, c). Remarkably, as pyrolysis-free synthesized catalysts, CPF-Fe/Ni achieved superior activity even for practical application to most reported overall water splitting electrocatalysts, and showed a high efficiency for transforming the electric power into chemical energy. The well-defined structure benefited from the mild synthesis route also offers an ideal platform to get insight into the structure-activity relationships during HER and OER process.

## Theoretical analyses

The addition of Ni generated spectacular promotion for the activity in EWS. To investigate the detailed mechanism and the effects of Ni, we employed computational hydrogen electrode (CHE) model to estimate the activity of OER and HER on CPF-Fe/Ni. According to the free energy curves (Fig. 5b), the OER activity on CPF-Fe was limited by the elementary reaction of *OOH → $O_2$ (ΔG = 2.04 eV), which was caused by the strong adsorption of intermediates. After the addition of Ni atoms, the adsorption of intermediates on CPF-Fe/Ni became weak with the ΔG of 1.81 eV, while the energy level was optimized. As calculated, CPF-Fe/Ni demonstrated lower overpotential ($\eta$ = 0.58 V) than that of pure CPF-Fe ($\eta$ = 0.81 V), indicating the higher OER activity. Meanwhile, for HER, CPF-Fe/Ni also displayed superior activity with optimized level (closer to zero). Since the weak adsorption for intermediate of OER and HER was the key factor for the high activity, the electronic structures of CPF-Fe/Ni and CPF-Fe were investigated to explore the physical mechanism of weak adsorption of intermediate (Fig. 5a). In the partial density of state (PDOS) of Fe-3d (Fig. 5c), the spin-up channel displayed obviously band gap, while the spin-down channel was conductive. For CPF-Fe/Ni, the peak at −2 ~ +1 eV was narrowed, especially for the peak at +0.6 eV which was sharp at spin-down channel. After the introduction of Ni atoms, the Fe-3d near the Fermi level became more

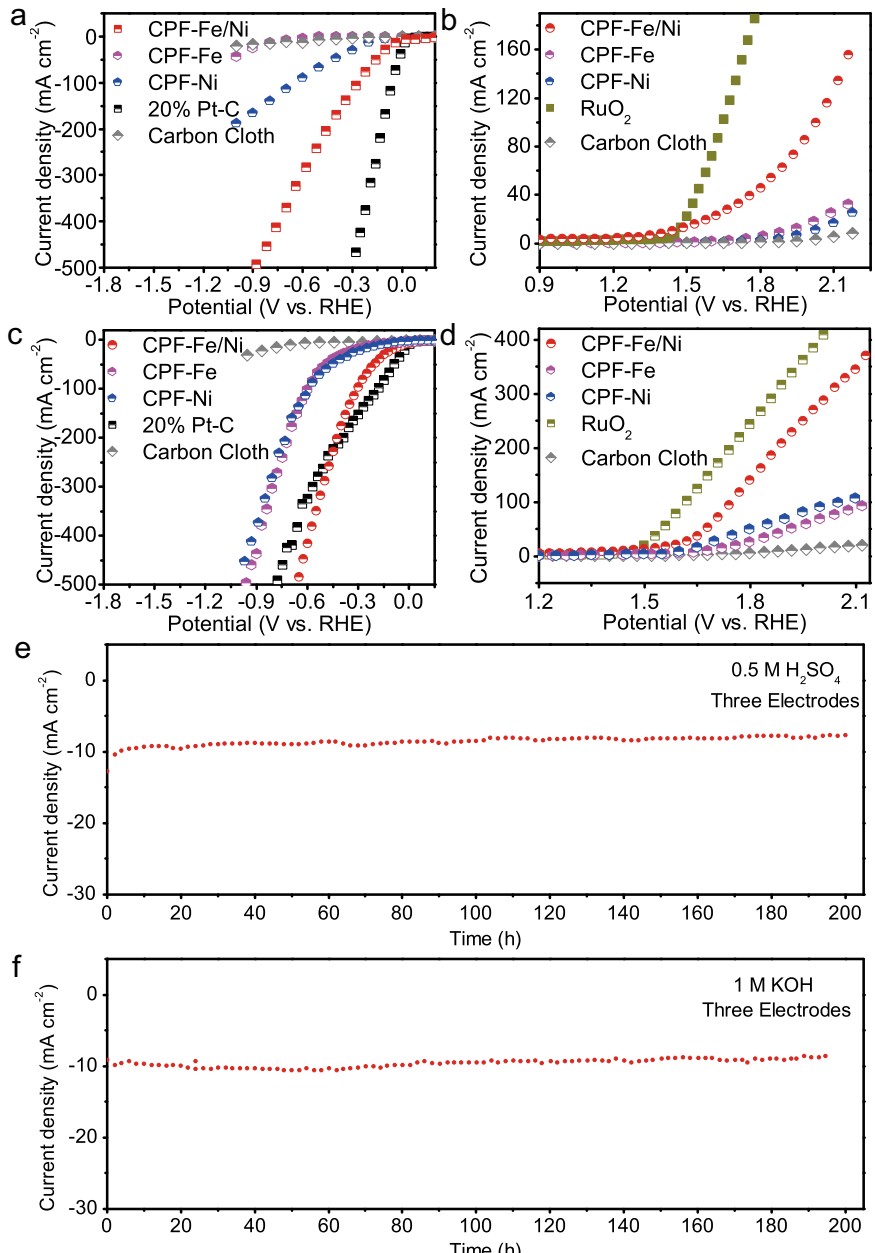

**Fig. 3 | HER and OER performance overall water-splitting performance of the CPF-Fe/Ni in 0.5 M H₂SO₄ and 1 M KOH.** a HER and b OER polarization curves with a speed of 10 mV s⁻¹ in 0.5 M H₂SO₄; c HER and d OER polarization curves with a speed of 10 mV s⁻¹ in 1 M KOH aqueous solution; The chronopotentiometry curves of CPF-Fe/Ni in 0.5 M H₂SO₄ (e) and 1 M KOH (f) at 10 mA cm⁻², The current of the cathode was collected to evaluate the catalytic activity and stability of CPF-Fe/Ni. All measurements were calibrated with *iR*-compensation.

local, leading to reduction of the electron transferring in the Fe-3d orbital. Thus, the adsorption would be weakened on CPF-Fe/Ni and the activity would be improved. Moreover, the differential charge results revealed that the electron transferring occurred between the Fe in different layers, while there was no interaction between Fe and Ni (Fig. 5d).

As revealed by the experimental results, the coordinated Fe and Ni sites were exclusively dispersed in the single-atom format and uniformly anchored throughout the surface in the HAADF STEM images (Fig. 1f). Besides, the synchrotron-based XANES and the extended X-ray absorption fine structure (EXAFS) spectra also confirmed that the existence of metal-N coordination and the absence of metal-metal bonds. The absence of WT singles located around ~5.2 Å⁻¹ that derived from Ni-metal bond and Fe-metal bond[39] further prove that no Ni-Fe

bond exits in CPF-Fe/Ni. Thus, combine experimental and theoretical calculations, it could be concluded that there was no interaction between Fe and Ni. Meanwhile, interactions of Fe in different layers could generate the electron transferring between the two layers, resulting strong adsorption of intermediate. The introduction of Ni would effectively weaken the adsorption of intermediate, which further promoted the reaction activity, the joint of Ni and Fe sites generated electronic structures, leading to excellent water-splitting catalytic activity of CPF-Fe/Ni.

## Discussion

Dual-function catalysts with high activity for both HER and OER play vital roles for EWS. Specially, catalysts that adapt to various pH values could benefit the efficient hydrogen production from water

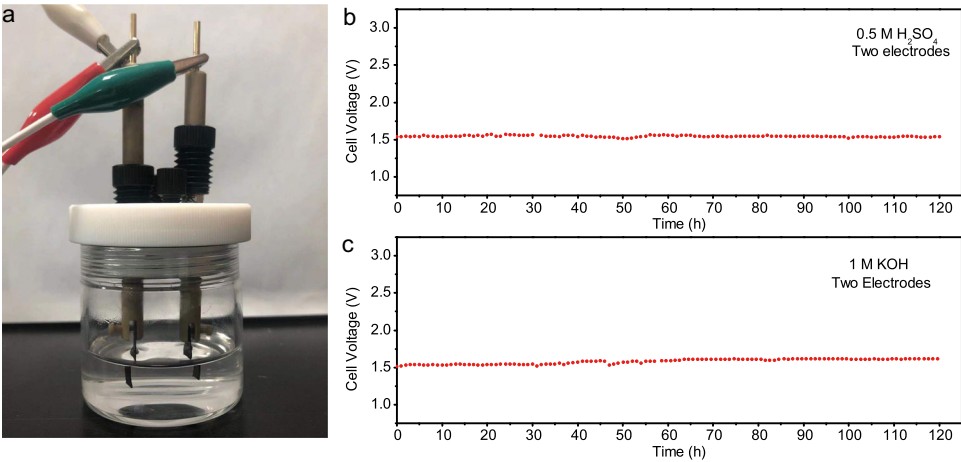

**Fig. 4 | Overall water-splitting performance. a** Digital photograph of the homemade two-electrode cell for water splitting. The corresponding chronopotentiometric curves at 10 mA cm⁻² in **b** 0.5 M H₂SO₄, and **c** 1 M KOH.

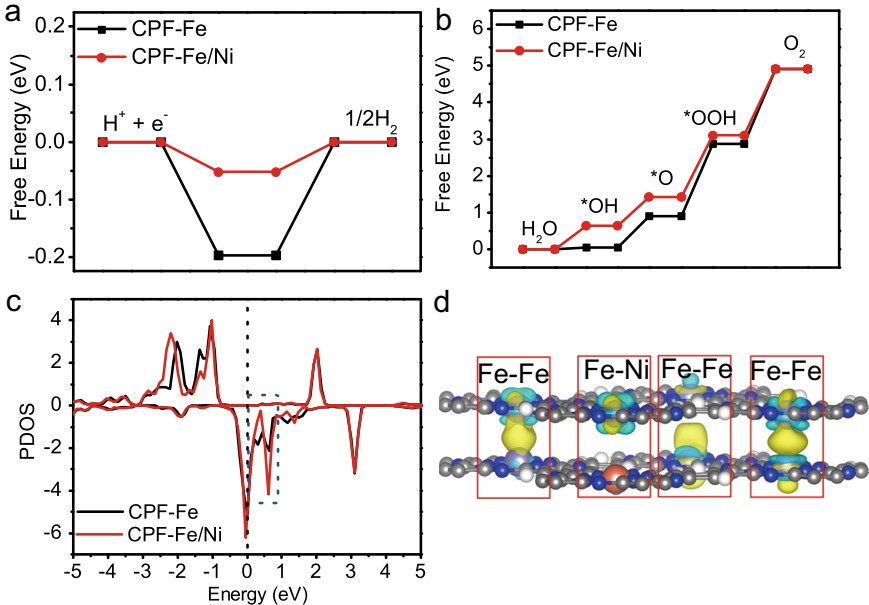

**Fig. 5 | Theoretical calculations.** The free energy curves of **a** HER and **b** OER on CPF-Fe and CPF-Fe/Ni, respectively; **c** The partial density of state for Fe-3d in CPF-Fe and CPF-Fe/Ni; **d** The differential charge for Fe-Fe and Fe-Ni site in CPF-Fe/Ni. The yellow and blue area denote the electron accumulation and loss. The isosurface is 0.002 e/Å³.

electrolysis. In our work, we developed pyrolysis-free synthetic route for single atomic catalysts and induced the bi-functional catalytic capability by adding heteroatoms. The as-prepared CPF-Fe/Ni exhibited considerable catalytic capability and stability for overall water splitting in both acidic and alkaline media. Along with simple synthesis process and low preparation cost, our work paved a promising avenue for efficient hydrogen production. More importantly, benefiting from the mild pyrolysis-free method, the structure and the active sites of catalysts could be well defined, providing ideal platforms to elucidate the catalytic process. Based on the as-designed CPF-Fe/Ni, we obtained in-depth sight into the mechanism. The introduction of Ni atoms reduced the electron transferring between the two layers and weakened the adsorption of intermediates, leading to optimized energy level and enhanced reaction activity. In summary, our work developed a route towards efficient EWS catalysts applicable for wide-pH range and successful showcase of model catalyst for mechanism studies.

## Methods

### Materials and reagents

All chemicals are analytical grade were purchased through commercial suppliers (Aladdin and Sigma-Aldrich), and used without further purification. Water used in this work was purified using the Milli-Q purification system.

### Experimental Details

**Synthesis of CPF-Fe.** Benzene-1,2,4,5-tetracarbonitrile (TCNB) was used as the monomers via a solvothermal process (180 °C for 4 days) with 1,8-Diazabicyclo(5,4,0)undec-7-ene (DBU) as the catalysts. Typically, TCNB (0.10 g, 0.560 mmol), ferric chloride (0.076 g 0.280 mmol), and DBU (0.10 mL) were dissolved in 9 mL ethylene glycol and DMF (v:v = 9:1), the reaction was going on 24 h. After cooling to room temperature, the precipitate was collected by filtration, and washed with water. The solid was dried to afford CPF-Fe as a black powder (Yield: 83%).

## Synthesis of CPF-Fe/Ni

CPF-Fe (0.10 g), $NiCl_2 \cdot 6H_2O$ (0.15 mg, 0.63 mmol), and KOH (3.75 g, 0.067 mmol) were mixed in 100 mL DMF, the reaction was going on 20 min under microwave (400 W). It has been reported that the CPF-Fe support could be well dispersed under alkaline conditions. Thus, KOH was used as the solvent for the reaction to promote the formation of bimetallic single-atom catalysts. After cooling to room temperature, the precipitate was collected by centrifugation, and washed with 3 M HCl and water. Then, the precipitate in black was dried under vacuum, to give CPF-Fe/Ni (Yield: 71%).

## Electrochemical measurements

Electrochemical measurements were carried out at ambient temperature and pressure with CHI 660e and OUTLAB. Electrocatalyst inks were prepared by dispersing 3 mg of catalyst into a solution containing 50 μL of 5% Nafion solution and 450 μL of DMF, followed by ultrasonication for 30 min. The mass loading of CPF-Fe/Ni on the electrode was ~192 μg cm$^{-2}$ with Fe and Ni single atoms around 22.8 μg cm$^{-2}$ and ~2.6 μg cm$^{-2}$, respectively. A three-electrode cell configuration was employed with a working electrode of carbon cloth (0.5 cm × 0.5 cm), a counter electrode of graphite rod and a Ag/AgCl as the reference electrode. Before each experiment, the carbon cloth was washed with $CH_3OH$ and $H_2O$. Then an aliquot of 10 μL of the catalyst ink was dropcasted on the carbon cloth and allowed to dry in air. All electrode potentials reported herein were converted to the RHE scale using

$$E(vs.RHE) = E(vs.SCE) + 0.197\,V \tag{1}$$

and

$$E(vs.RHE) = E(vs.SCE) + 1.023\,V \tag{2}$$

for the measurements in acidic and alkaline media, respectively. The zero point of RHE was determined by the equilibrium potential of HER/HOR using Pt/C as working electrode in $H_2$-saturated electrolyte. All polarization curves were corrected for the iR compensation (the specific percentage of the correction is 100%). The overpotential $\eta$ was calculated by

$$\eta = E(vs.RHE)\,V \tag{3}$$

for HER and

$$\eta = E(vs.RHE) - 1.23\,V \tag{4}$$

for OER, respectively. The scan speed was 10 mV s$^{-1}$.

## Turnover frequency (TOF) calculations

TOF values of difffferent electrodes were calculated based on the equation:

$$TOF = (J \times A)/(4 \times F \times n) \tag{5}$$

where $J$ (mA cm$^{-2}$) is the current density at the overpotential of 400 mV for OER and 200 mV for HER; A is the surface area of electrode; F is the Faraday constant (96,485 C/mol); $n$ is molar number of active sites on the electrode[37].

## Computational Details

The Vienna ab-initio simulation package (VASP)[40–44] is employed to perform the spin-polarized density functional theory (DFT) calculations in this work. The core electrons of atoms were treated by Blöchl's all-electron-like projector augmented wave (PAW) method[44,45]. The Perdew-Burke-Ernzerbof (PBE) within the generalized gradient approximation functional (GGA) is adopted to describe the exchange and correlation effects[46]. The plane wave energy cutoff was set as 400 eV. The Gaussian scheme was employed for electron occupancy with an energy smearing of 0.1 eV. The first Brillouin zone was sampled in the Monkhorst–Pack grid[47]. The 3 × 3 × 1 k-point mesh is used for the calculations. The energy (converged to 1.0 ×10$^{-6}$ eV/atom) and force (converged to 0.01 eV/Å) were set as the convergence criterion for geometry optimization.

The free energy change of the elementary reaction in OER is estimated by the following expression:[48,49]

$$\Delta G = \Delta E + \Delta ZEP - T\Delta S + \Delta G_U + \Delta G_{pH} + \Delta G_{field} \tag{6}$$

where the total electron energy change between initial and final states of the reaction obtained by DFT computations was regarded as $\Delta E$. $\Delta ZPE$ and $\Delta S$ represents the zero-point energy changes and the entropy change, respectively. $T$ means that the reaction temperature (298.15 K). $\Delta G = -eU$, where $e$ and $U$ represents the electrode potential relative to the standard hydrogen electrode and the transferred charge, respectively. In aqueous solution environments, the effect of $pH$ on free energy has also been considered, according to:

$$\Delta G_{pH} = k_B T\,ln\,10 \times pH \tag{7}$$

where $k_B$ is the Boltzmann constant, and $pH = 0$ for acid medium[48,50]. The free energy correction $\Delta G_{field}$ has been neglected because of the electrochemical double layer[48,51]. Gas-phase $H_2O$ at 0.035 bar was used as the equilibrium point of the gas-phase and liquid $H_2O$ at 298.15 K. The free energy of $O_2$ is assessed via the reaction $2H_2O \rightarrow O_2 + 2H_2$, with the free energy values of 4.92 eV at 298.15 K and 0.035 bar[48]. According to the hydrogen electrode model proposed by Nørskov et al, the free energy of $(H^+ + e^-)$ in solution at standard conditions ($pH = 0$, $U = 0$) is equal to that of $1/2H_2$[48]. The entropy of the $H_2$ is obtained from the NIST database[52], and the entropies of the OER intermediates were obtained from the frequencies calculations.

The working potential is the potential that keeps all the elementary reactions to be exothermic. It is defined as:

$$U_{OER} = max[\Delta G_x]/ne \tag{8}$$

where $n$ is the number of electrons transferred for each reaction, and $e$ is the elementary charge[53,54]. And the overpotential is calculated by

$$\eta = U_{OER} - 1.23\,V \tag{9}$$

The HER is also estimated by the Gibbs free-energy change ($\Delta G_{ads}$) of H:

$$\Delta G_{ads} = \Delta E_{ads} + \Delta E_{ZPE} + 0.20\,eV \tag{10}$$

The $\Delta E_{ads}$ is defined as follows:

$$\Delta E_{ads} = E_{H/slab} - (E_{slab} + 1/2E_{H2}) \tag{11}$$

where the $E_{H/slab}$ is the total energy of H atom on CPF-FeFe/-FeNi, $E_{slab}$ is the total energy of the CPF-FeFe/-FeNi and $E_H$ is the energy of H atom reference to the gas $H_2$. The first two terms are calculated with the same parameters. The third term is calculated by setting the isolated $H_2$ in a box of 12 Å × 12 Å × 12 Å.

## Model

To meet the experimental configuration, the double layers of CPF-Fe (d-CPF-Fe) are built. A (2 × 2) supercell of d-CPF-Fe is employed, while a vacuum layer of 20 Å was employed along the c axis to avoid the periodic interaction. For Ni doped, one of Fe will be replaced by Ni atom.

## Data availability

All data needed to evaluate the conclusions in the paper are present in the paper and/or the Supplementary Materials. Source data are provided with this paper.

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

## Acknowledgements

This work was supported by the National Natural Science Foundation of China (No. 22005273, 92061201, 22201084), the National Science Fund for Distinguished Young Scholars (21825106, 22225109). We thank J.H.H., J.W. for helpful discussion and suggestions; K.L. for the analysis of the DFT result.

## Author contributions

P.P. and S.Q.Z. supervised and led this project. Y.Z. performed synthesis, structural characterizations, and electrochemical tests. D.Q.L. and K.W. helped prepare the materials and structural characterizations. Y.Z. and P.P. co-wrote this paper. P.P. analyzed DFT calculations. B.L. and Y.Q.L. discussed the manuscript. All authors provided critical feedback, helped shape the research and manuscript, and commented on the manuscript.

## Competing interests

The authors declare no competing interests.
