## [Peer Review File · Nature Communications]

A pyrolysis-free Ni/Fe bimetallic electrocatalyst for overall water splittingREVIEWER COMMENTS

Reviewer #1 (Remarks to the Author):

The authors present an interesting work on a two-dimensional conjugated phthalocyanine framework (CPF) simultaneously contained single atomic Ni and Fe sites with improved activity towards overall water splitting in wide-pH range. It is noteworthy that the CPF-Fe/Ni exhibited excellent stability for OER in acidic media, which is always considered to require the use of noble metal catalysts such as Ir or Ru. The work would be of broad interest to the research community. I'd be happy to recommend acceptance of this study after the authors address the following points:

1. The authors are right in saying that "Furthermore, considering that the partially embedded nanoparticle active sites would result in a reduction of the catalytically active surface, various routes have been developed to improve the atomic utilization over the past decades. For example, derivatives of three-dimensional MOFs (Metal utilization rate) with open nanostructures can effectively improve atomic utilization rate." However, there are still other ways to solve the problem. Authors are encouraged to refer to studies others than those they cited. See, for instance, some three-dimensional open nano-netcage electrocatalysts, *Nat Commun* 10, 4875 (2019).
2. In the Figure 1f, "In the aberration corrected images it could be found that the coordinated Fe and Ni sites were exclusively dispersed in the single-atom format and uniformly anchored throughout the surface." Frankly, I can't see it clearly.
3. In the Figure 2a and c, why is there no 1s-4pz characteristic peak of M-N₄ configuration in the XANES spectra of Fe and Ni. The authors need to explain it, or give the corresponding references.
4. It's unclear if the electrochemical results have been compensated with solution resistance in the manuscript, and I suggest the Tafel analysis use the compensated data. When using the Tafel slopes to discuss the reaction kinetics, the author needs to select a region with a relatively small current density to ensure that it is in the kinetics region.
5. In the "Electrochemical measurements" section, "All electrode potentials reported herein were converted to the RHE scale using $E(\text{vs. RHE}) = E(\text{vs. SCE}) + 0.197 \text{ V}$ and $E(\text{vs. RHE}) = E(\text{vs. SCE}) + 1.023 \text{ V}$ for the measurements in acidic and alkaline media, respectively". To ensure the authenticity of electrochemical data, the zero point of RHE needs to be calibrated using the HER/HOR equilibrium potential using Pt/C catalyst in H₂-saturated solutions.
6. The authors claim that "Moreover, the differential charge results revealed that the electron transferring occurred between the Fe in different layers, while there was no interaction between Fe and Ni (Fig. 5d)." In my opinion, the authors need to give further experimental evidence to explain the high activity of CPF-Fe/Ni.
7. Figure 3c and f – the chronopotentiometry curves – very stable. Please specify the corresponding reactions, HER or OER. Similar problems – Supplementary Figure 10 and 11, please specify the corresponding reactions.
8. Supporting Information Table 2 – ICP of CPF-Fe/Ni before and after catalysis – Please specify the corresponding solutions, acid or bases. Of course, in best, authors should give the all ICP results after catalysis in acid and bases, respectively.
9. In the title of the article, the authors claimed that "Pyrolysis-free synthesized single-atom

electrocatalysts with bimetallic sites for high-performance overall water splitting in wide-pH range." – Authors should give the more performance tests in other pHs, especially neutral solutions.

Reviewer #2 (Remarks to the Author):

Zang and collaborators developed a bi-function catalyst with single atomic active sites through pyrolysis-free route, the as-prepared catalyst showed an out-standing catalytic capability as a bifunctional HER and OER pH-universal catalyst. This work reported an interesting catalytic performance for overall water splitting based on Fe/Ni active sites. According to the results, the overpotential for HER/OER were 23/201 mV and 42/194 mV in 0.5 M H₂SO₄ and 1 M KOH, respectively. In addition, the catalyst also exhibited considerable stability. The authors also performed the theoretical calculations in detail. Overall, these findings are impressive and well presented and organized. This work is also a successful showcase for new synthetic routes of single atomic catalysts. Thus, this work is acceptable for Nat. Commun. Here are some minor concerns which should be carefully addressed before the paper can be published:

With regard to the synthesis of CPF-Fe/Ni material, CPF-Fe was synthesized by a pyrolysis-free route firstly. The authors should add more discussion for the possible special characteristic of such synthesized materials compared with the Fe single atoms based materials by pyrolysis route.

The reported CPF-Fe/Ni was subsequently prepared via a post-modification route, in which the ion exchange reaction was continuously triggered under a microwave reaction to generate the bimetallic conjugation of CPF-Fe/Ni. Compared with the catalysts prepared by one-pot method or in situ substitution by a heterometal ion, are there some unique features of CPF-Fe/Ni as a bifunctional catalysts for HER and OER?

The authors have performed lots of tests to study the stability of the catalysts and compared such as FTIR, TEM, HAADF STEM before/after the catalysis process to verify the stability. The authors should also include the comparison of XPS tests before/after the catalysis process to further confirm the stability of elemental state. Thus, XPS data tested under the same procedure after catalysis process should be added.

Some issues about the figures should revised. For example, the colors of hydrogen and oxygen are contrary in the note of supplementary Figure 19-21. The format should be uniform both in the manuscript and the supporting information, such as "SupplementaryFigure 2." should be "Supplementary Figure 2". The authors should examine carefully.

The title of the directory does not match the corresponding picture below in supporting information. For example, in the part of "Supplementary Text", "Supplementary Figure 6. Tafel slopes for 20% Pt-C, CPF-Fe, CPF-Ni and CPF-Fe/Ni in 1 M KOH" should be corrected as "Supplementary Figure 6. Tafel slopes for 20% Pt-C, CPF-Fe, CPF-Ni and CPF-Fe/Ni as HER catalyst in 1 M KOH". There are also some typos existed.

Some grammar and format mistakes should be corrected. The authors are suggested to check the manuscript as well as the supporting information.

Reviewer #3 (Remarks to the Author):

Comments

This work developed the single-atom electrocatalyst with bimetallic sites, which was highly efficient for overall water splitting in wide-pH range. Impressively, this catalyst achieved demonstrate low overpotential with high stability for overall water splitting in both acidic and alkaline electrolytes. The authors have conducted substantial characterizations to verify their outcomes. The research has been well performed and the results are convincing. The pyrolysis-free synthesis for the preparation of highly efficient Fe and Ni based catalysts holds valuable novelty, which also is a good showcase for the catalytic process studies. Therefore, I would like to recommend the publication in Nature Communication. Following are some minor issues which could be addressed before final publication:

1. The authors have claimed that “the pyrolysis-free route was mild and ensure the well-defined structure and precise location of active centers, offering a new path for further explanation of the mechanism”. Since such synthesis for single atomic catalysts is the important novelty of this work, the authors should provide more description or discussion for pyrolysis-free.
2. According to the results reported by the authors, it could be concluded that the presence of Ni was very important or even essential for the catalytic performance during the overall water splitting. Thus, is it possible to supply more study about the control of Ni species during the synthesis? For example, to explore whether the Ni content can be affected by the reaction conditions.
3. As the authored reported, Ni single atoms were introduced by a post-synthesized. During this process, it could be noted that KOH was used in the synthesis. What the role did KOH play, and how to avoid Ni(OH)₂ was produced?
4. This work involved a large number of literature comparisons, which was very necessary. These comparisons need to be guaranteed under the same conditions. The authors are suggested to clearly mark the comparison conditions.
5. Given to the scientificity and the continuity of the data, the video of overall water splitting process is more intuitive and easy to accept. The authors are suggested to provide the videos of the water splitting.
6. Some typos and errors in grammar and format need to be corrected: On Page 3, Line 9, “have attracted numerous attentions” should be corrected for “have attracted numerous attention”. On Page 4, Line 11, “various routes have been developed to improve the atomic utilization over the past decades.” should be corrected for “various routes had been developed to improve the atomic utilization over the past decades”. Similar mistake also found on Page 5, Line 7 and 8. The format of references should be unified. Such as in Ref 32 “Energy Environ. Sci., 15” should be “Energy Environ. Sci. 15”. Meanwhile, in Ref 10 “CO₂” should be “CO₂”.

Point by Point Response

Reviewer #1

The authors present an interesting work on a two-dimensional conjugated phthalocyanine framework (CPF) simultaneously contained single atomic Ni and Fe sites with improved activity towards overall water splitting in wide-pH range. It is noteworthy that the CPF-Fe/Ni exhibited excellent stability for OER in acidic media, which is always considered to require the use of noble metal catalysts such as Ir or Ru. The work would be of broad interest to the research community. I'd be happy to recommend acceptance of this study after the authors address the following points:

Response: Thanks so much for the comments.

1. *The authors are right in saying that "Furthermore, considering that the partially embedded nanoparticle active sites would result in a reduction of the catalytically active surface, various routes have been developed to improve the atomic utilization over the past decades. For example, derivatives of three-dimensional MOFs (Metal utilization rate) with open nanostructures can effectively improve atomic utilization rate." However, there are still other ways to solve the problem. Authors are encouraged to refer to studies others than those they cited. See, for instance, some three-dimensional open nano-netcage electrocatalysts, Nat Commun 10, 4875 (2019).*

Response: Many thanks for the advice. There are some other ways to solve the mentioned problem including the three-dimensional open nano-netcage electrocatalysts and it is very necessary to refer these studies in order for a better understanding. According to the reviewer's suggestion, we performed a more detailed literature research and discussion about the ways to improve atomic utilization. All the changes are highlighted in yellow background:

"For example, derivatives of three-dimensional MOFs (Metal-Organic Frameworks) with open nanostructures can provide a frameworks with large area for catalysis. It has been developed that three-dimensional open nano-netcage derived electrocatalysts could effectively improve atomic utilization rate. Nowadays, many porous materials such as nanosheets with rough surface and expanding layers,¹⁷ three-dimensional open nano-netcage electrocatalysts²⁴ and nanotube structures with well-defined inner channels and a large surface area²⁵ can effectively improve atomic utilization rate.

References:

17. Zhou J., Dou Y., Wu X.-Q., Zhou A., Shu L., Li J.-R.. Alkali-Etched Ni(II)-Based Metal-Organic Framework Nanosheet Arrays for Electrocatalytic Overall Water Splitting. *Small* **16**, 1906564 (2020).

24. Zhuang Z., *et al.* Three-dimensional open nano-netcage electrocatalysts for efficient pH-universal overall water splitting. *Nat. Commun.* **10**, 4875 (2019).

25. Pan Y., *et al.* Core-Shell ZIF-8@ZIF-67-Derived CoP Nanoparticle-Embedded N-Doped Carbon Nanotube Hollow Polyhedron for Efficient Overall Water Splitting. *J. Am. Chem. Soc.* **140**, 2610-2618 (2018).

2. In the Figure 1f, "In the aberration corrected images it could be found that the coordinated Fe and Ni sites were exclusively dispersed in the single-atom format and uniformly anchored throughout the surface." Frankly, I can't see it clearly.

Response: Many thanks for the Reviewer's suggestion. To this concern, we have uploaded a new figure with higher resolution in the revised version for more clearly observation. The changes are highlighted in yellow background:

Fig. 1 Schematic illustration and structural characterization of CPF-Fe/Ni. (a) The synthesis process of CPF-Fe/Ni, (b) Solid-state CP-MAS ¹³C-NMR spectra of CPF-Fe and CPF-Fe/Ni, (c) High-resolution Fe2p spectra of CPF-Fe and CPF-Fe/Ni, the black curves are the fits data while the gray lines are the experimental results, (d) SEM and (e) TEM image of CPF-Fe/Ni, (f-i) HAADF STEM image and EDS-mapping results of CPF-Fe/Ni.

3. In the Figure 2a and c, why is there no 1s-4pz characteristic peak of M-N4 configuration in the XANES spectra of Fe and Ni. The authors need to explain it, or give the corresponding references.

Response: Thanks for the professional suggestion. Considering the reviewer's concerns, we have investigated more literatures about XANES spectra of M-N4 configuration and discovered many similar results as our results. According to the suggestions and for a better understanding, we have added the corresponding references and the discussions. Typically, the pre-edge peak is due to a 1s-4pz shakedown transition characteristic for a square-planar configuration with high D4h

symmetry. It should be noted that the pre-edge profile of CPF-Fe/Ni is similar to that of FePc and NiPc, but exhibits a slight difference in intensities, suggesting that the Fe/Ni center is coordinated with four N atoms by achieving a square-planar Fe/Ni-N₄ molecular structure. (*Nat. Catal.* **1**, 63-72 (2018); *Nat. Catal.* **2**, 259-268 (2019); *Adv. Mater.* **31**, 1903955 (2019); *Angew. Chem. Int. Ed.* **58**, 10677-10682 (2019); *Adv. Mater.* **32**, 2004670 (2020)). In order to make the image information clearer, we enlarged the local area (inserted images) and provided a more delated discussion about the XANES spectra in the revised manuscript. All the changes have been highlighted in yellow background as followed:

Fig. 2 The synchrotron-based XANES and the extended X-ray absorption fine structure (EXAFS) spectra. (a) Normalized XANES and (b) radial structure functions (RSFs) of the Fe K-edge obtained by Fourier transformation k^3 -weighted EXAFS results, with the Fe foil, Fe₂O₃, Fe₃O₄, and FePc serving as references; (c) Normalized XANES and (d) radial structure functions (RSFs) of the Ni K-edge obtained by Fourier transformation k^3 -weighted EXAFS results, with the Ni foil, Ni(OH)₂, NiO, and NiPc serving as references.

According to previous reports, the pre-edge peak could be attributed to a 1s-4p_z shakedown transition characteristic for a square-planar configuration with high D_{4h} symmetry.^{32,33} In this work, it should be noted that the pre-edge profile of CPF-Fe/Ni is similar to that of iron phthalocyanine (FePc) and nickel phthalocyanine (NiPc) (Fig. 2a and c), but exhibits a slight difference in intensities. These phenomenons suggested that the Fe/Ni center is coordinated with four N atoms by achieving a square-planar Fe/Ni-N₄ molecular structure.³⁴⁻³⁶

References:

32. Fei H., *et al.* General synthesis and definitive structural identification of MN₄C₄ single-atom catalysts with tunable electrocatalytic activities. *Nat. Catal.* **1**, 63-72 (2018).

33. Zhang L., *et al.* Single Nickel Atoms on Nitrogen-Doped Graphene Enabling Enhanced Kinetics of Lithium–Sulfur Batteries. *Adv. Mater.* **31**, 1903955 (2019).

34. Zhong H., *et al.* A Phthalocyanine-Based Layered Two-Dimensional Conjugated Metal-Organic Framework as a Highly Efficient Electrocatalyst for the Oxygen Reduction Reaction. *Angew. Chem. Int. Ed.* **58**, 10677-10682 (2019).

35. Zhu Z., *et al.* Coexisting Single-Atomic Fe and Ni Sites on Hierarchically Ordered Porous Carbon as a Highly Efficient ORR Electrocatalyst. *Adv. Mater.* **32**, 2004670 (2020).

36. Wan X., *et al.* Fe-N-C electrocatalyst with dense active sites and efficient mass transport for high-performance proton exchange membrane fuel cells. *Nat. Catal.* **2**, 259-268 (2019).

4. *It's unclear if the electrochemical results have been compensated with solution resistance in the manuscript, and I suggest the Tafel analysis use the compensated data. When using the tafel slopes to discuss the reaction kinetics, the author needs to select a region with a relatively small current density to ensure that it is in the kinetics region.*

Response: Many thanks for the professional advice. As the reviewer mentioned, compensating data with solution resistance through *iR*-compensation is very important in the testing process and for the related analysis. It is one of the essential requirements for the electrochemical tests. In our work, all of the measurements and the as-obtained data were calibrated with *iR*-compensation including the Tafel analysis. When calculating the Tafel slope, the minimum current density region that meets kinetics region the required has been used. We are sorry that we did not clearly indicate in the test conditions. We emphasized the related parts in the revised manuscript and added the information in the revised experimental section. All the changes have been highlighted in yellow background as followed:

All polarization curves were corrected for the *iR* compensation (the specific percentage of the correction is 100%).

5. *In the "Electrochemical measurements" section," All electrode potentials reported herein were converted to the RHE scale using $E(\text{vs. RHE}) = E(\text{vs. SCE}) + 0.197 \text{ V}$ and $E(\text{vs. RHE}) = E(\text{vs. SCE}) + 1.023 \text{ V}$ for the measurements in acidic and alkaline media, respectively". To ensure the authenticity of electrochemical data, the zero point of RHE needs to be calibrated using the HER/HOR equilibrium potential using Pt/C catalyst in H_2 -saturated solutions.*

Response: Thanks for the professional suggestions. As the reviewer mentioned, it is very important and essential to calibrate the zero point of RHE. During the tests, we have calibrated the RHE by the equilibrium potential of HER/HOR using Pt/C catalyst in H_2 -saturated solutions. We are very sorry that we missed the statement and made it unclearly. In case of any misunderstandings, we added this very important information in the revised manuscript. All the changes have been highlighted in yellow background as followed:

...for the measurements in acidic and alkaline media, respectively. The zero point of RHE was determined by the equilibrium potential of HER/HOR using Pt/C as working electrode in H₂-saturated electrolyte.

6. The authors claim that "Moreover, the differential charge results revealed that the electron transferring occurred between the Fe in different layers, while there was no interaction between Fe and Ni (Fig. 5d)." In my opinion, the authors need to give further experimental evidence to explain the high activity of CPF-Fe/Ni.

Response: Thanks so much for the professional advice. In case for the concern, we have added more discussion based on the experimental results to further explain the interaction between Fe and Ni, as well as the relate activity of CPF-Fe/Ni. The changes are highlighted in yellow background as followed:

Moreover, the differential charge results revealed that the electron transferring occurred between the Fe in different layers, while there was no interaction between Fe and Ni (Fig. 5d). As revealed by the experimental results, the coordinated Fe and Ni sites were exclusively dispersed in the single-atom format and uniformly anchored throughout the surface in the HAADF STEM images (Figure 1f). Besides, the synchrotron-based XANES and the extended X-ray absorption fine structure (EXAFS) spectra also confirmed that the existence of metal-N coordination and the absence of metal-metal bonds. The absence of WT singles located around $\sim 5.2 \text{ \AA}^{-1}$ that derived from Ni-metal bond and Fe-metal bond³⁹ further prove that no Ni-Fe bond exists in CPF-Fe/Ni. Thus, combine experimental and theoretical calculations, it could be concluded that there was no interaction between Fe and Ni. Meanwhile, interactions of Fe in different layers could generate the electron compensation on the surface Fe atoms, resulting strong adsorption of intermediate. The introduction of Ni would effectively eliminate the electron compensation and weaken the adsorption of intermediate, which further promoted the reaction activity, the joint of Ni and Fe sites generated unique electronic structures, leading to excellent water splitting catalytic activity of CPF-Fe/Ni.

39. Zeng Z., *et al.* Orbital coupling of hetero-diatomic nickel-iron site for bifunctional electrocatalysis of CO₂ reduction and oxygen evolution. *Nat. Commun.* **12**, 4088 (2021).

7. Figure 3c and f – the chronopotentiometry curves – very stable. Please specify the corresponding reactions, HER or OER. Similar problems – Supplementary Figure 10 and 11, please specify the corresponding reactions.

Response: Many thanks for the careful advices. We have specified the discussion of the concerned parts and added more detailed description in case of any misleading. The changes have been highlighted in yellow background in the revised version as followed:

“Then, a three-electrode cell configuration was employed with Ag/AgCl as the reference electrode. Both the working electrode and the counter electrode are drop-coated with carbon cloth of CPF-Fe/Ni. During the catalysis reaction, the cathode and anode produced hydrogen evolution reaction and oxygen evolution reaction, respectively. The current of the cathode was collected to evaluate the catalytic activity and stability of CPF-Fe/Ni as hydrogen evolution and oxygen evolution bifunctional catalysts.”

Fig. 3... The current of the cathode was collected to evaluate the catalytic activity and stability of CPF-Fe/Ni...

8. Supporting Information Table 2 – ICP of CPF-Fe/Ni before and after catalysis – Please specify the corresponding solutions, acid or bases. Of course, in best, authors should give the all ICP results after catalysis in acid and bases, respectively.

Response: Many thanks for the advice. We have updated the data in this section with the mentioned conditions including solution and concentration. The new table have been added in the revised Supporting Information and highlighted in yellow background as followed:

Supplementary Table 2. ICP of CPF-Fe/Ni before and after catalysis

	Fe (%)	Ni (%)
CPF-Fe/Ni before catalysis	11.90	1.35
CPF-Fe/Ni after catalysis in 0.5 M H ₂ SO ₄	11.16	1.21
CPF-Fe/Ni after catalysis in 1 M KOH	11.32	1.18

9. In the title of the article, the authors claimed that "Pyrolysis-free synthesized single-atom electrocatalysts with bimetallic sites for high-performance overall water splitting in wide-pH range." – Authors should give the more performance tests in other pHs, especially neutral solutions.

Response: Many thanks for the advice. As suggested, we measured the HER and OER polarization curves based on CPF-Fe/Ni under more pH conditions (0.5 M H₂SO₄, 0.05 M H₂SO₄, 1M KCl, 0.01 M KOH and 1 M KOH). Due to the low concentration of adsorbed reactants on the catalyst surface, the reaction kinetics in the neutral electrolyte are extremely slow. In other pH, CPF-Fe/Ni has a highly effective catalytic activity on the overall water splitting. We have added these supplement data in the revised Supporting Information. The changes have been highlighted in yellow background as followed:

Even the reaction kinetics in the neutral electrolyte (1 M KCl) are slow due to the low concentration of adsorbed reactants, however, as HER catalysts the CPF-Fe/Ni demonstrates very small over-potential of 23, 76, 42, 147 mV in 0.5 M H₂SO₄, 0.05 M H₂SO₄, 0.01 M KOH and 1 M KOH, respectively. Meanwhile, as OER catalysts

the over-potential of 201, 345, 310, 194 mV was obtained in 0.5 M H₂SO₄, 0.05 M H₂SO₄, 0.01 M KOH and 1 M KOH, respectively. Thus, these findings suggested the high efficiency of CPF-Fe/Ni as catalysts for the overall water splitting in wide-pH range.

Supplementary Figure 8. HER polarization curves with a speed of 10 mV s⁻¹ in a wide pH.

Supplementary Figure 9. OER polarization curve with a speed of 10 mV s⁻¹ in a wide pH.

Reviewer #2

Zang and collaborators developed a bi-function catalyst with single atomic active sites through pyrolysis-free route, the as-prepared catalyst showed an out-standing catalytic capability as a bifunctional HER and OER pH-universal catalyst. This work reported an interesting catalytic performance for overall water splitting based on Fe/Ni active sites. According to the results, the overpotential for HER/OER were 23/201 mV and 42/194 mV in 0.5 M H₂SO₄ and 1 M KOH, respectively. In addition, the catalyst also exhibited considerable stability. The authors also performed the theoretical calculations in detail. Overall, these findings are impressive and well presented and organized. This work is also a successful showcase for new synthetic routes of single atomic catalysts. Thus, this work is acceptable for Nat. Commun. Here are some minor concerns which should be carefully addressed before the paper can be published:

Response: Thanks so much for the comments.

With regard to the synthesis of CPF-Fe/Ni material, CPF-Fe was synthesized by a pyrolysis-free route firstly. The authors should add more discussion for the possible special characteristic of such synthesized materials compared with the Fe single atoms based materials by pyrolysis route.

Response: Many thanks for the suggestions. In the revised version, we have added a depth discussion according to the suggestion. The changes have been highlighted in yellow background as followed:

Pyrolysis indeed has been very widely used in the past decades. Unfortunately, pyrolysis process is always carried out in specific atmosphere with high temperature, which are very energy-intensive. Meanwhile, the isolated metal atoms on the precursors are very unstable and tend to agglomerate during the pyrolysis process, not only reducing the efficient utilization of metal active sites but also making it difficult to realize the precise control. Thus, the development of non-carbonized synthesis strategies for the preparation of efficient electro-catalysts is challenging and vital. It could be expected that the pyrolysis-free route would lead to precisely synthetic control and a better maintained structure, holding the potentials to elucidate structure-function relationships for the guidance of optimum electro-catalysts.

The reported CPF-Fe/Ni was subsequently prepared via a post-modification route, in which the ion exchange reaction was continuously triggered under a microwave reaction to generate the bimetallic conjugation of CPF-Fe/Ni. Compared with the catalysts prepared by one-pot method or in situ substitution by a heterometal ion, are there some unique features of CPF-Fe/Ni as a bifunctional catalysts for HER and OER?

Response: Thanks for the comments. In this work, the bimetallic configuration of CPF-Fe/Ni was synthesized via a post-modification under a microwave reaction.

Compared with commonly used pyrolysis or one-pot solvothermal method, the synthesis route is more mild, easy to perform and conducive to the controllability of the synthesis process. This method could not only effectively reduce the formation of agglomeration, but also maintain the integrity of the structure to the greatest extent since the drastic treatments such as etching process were avoided. Moreover, the reactions are very simple, which hold the promising potentials for large-scale production. We have added the comparison in the revised version and marked in yellow background as followed:

... simultaneously contained single atomic Ni/N/C and Fe/N/C (termed as CPF-Fe/Ni). The mild synthetic route not only ensured the effectively reduce the formation of agglomeration, but also maintain the integrity of the structure.

The authors have performed lots of tests to study the stability of the catalysts and compared such as FTIR, TEM, HAADF STEM before/after the catalysis process to verify the stability. The authors should also include the comparison of XPS tests before/after the catalysis process to further confirm the stability of elemental state. Thus, XPS data tested under the same procedure after catalysis process should be added.

Response: Many thanks for the professional suggestions. According to the suggestion, we have collected the XPS results after the catalysis process both in acid and alkaline conditions, which showed a highly consistent with other measurement and further confirmed the stability of elemental state. The corresponding results have been added in the Supporting Information as Supplementary Figure 18 and 19. The changes of the relative parts in the manuscript are as followed. All the changes are highlighted in yellow background.

High-resolution N1s and Fe2p spectra of CPF-Fe/Ni after electrocatalysis testing in 0.5 M H₂SO₄ and 1 M KOH were collected (Supplementary Fig. 18 and 19), which also showed a highly consistent with other measurement and further confirmed the stability of elemental state.

Supplementary Figure 18. High-resolution N1s spectra of CPF-Fe/Ni after electrocatalysis testing in 0.5 M H₂SO₄ and 1 M KOH, respectively.

Supplementary Figure 19. High-resolution Fe2p spectra of CPF-Fe/Ni after electrocatalysis testing in 0.5 M H₂SO₄ and 1 M KOH, respectively.

Some issues about the figures should be revised. For example, the colors of hydrogen and oxygen are contrary in the note of supplementary Figure 19-21. The format should be uniform both in the manuscript and the supporting information, such as “SupplementaryFigure 2.” should be “Supplementary Figure 2”. The authors should examine carefully.

Response: Many thanks for the careful advice. We have corrected the mentioned concerns in the revised version in case of any misleading. The colors of hydrogen and oxygen in the note of supplementary Figure 21-23 have been corrected as “Hydrogen (green) and oxygen (red) generated at 0, 1.12, 2.23, 3.43, 4.58, 5.77, 7, 8.25, 9.5, and 10.83 min in 0.5 M H₂SO₄, respectively” and “Hydrogen (green) and oxygen (red) generated at 0, 3.55, 6.1, 8.45, 10.76, 13.02, 15.42, 17.67, 20.03, and 22.38 min in 1 M KOH, respectively.” The format in the Supplementary Text have been united. All the changes are highlighted in yellow background as followed:

Supplementary Figure 21. Hydrogen (green) and oxygen (red) generated at 0, 1.12, 2.23, 3.43, 4.58, 5.77, 7, 8.25, 9.5, and 10.83 min in 0.5 M H₂SO₄, respectively.

Supplementary Figure 23. Hydrogen (green) and oxygen (red) generated at 0, 3.55, 6.1, 8.45, 10.76, 13.02, 15.42, 17.67, 20.03, and 22.38 min in 1 M KOH, respectively.

The title of the directory does not match the corresponding picture below in supporting information. For example, in the part of “Supplementary Text”, “Supplementary Figure 6. Tafel slopes for 20% Pt-C, CPF-Fe, CPF-Ni and CPF-Fe/Ni in 1 M KOH” should be corrected as “Supplementary Figure 6. Tafel slopes for 20% Pt-C, CPF-Fe, CPF-Ni and CPF-Fe/Ni as HER catalyst in 1 M KOH”. There are also some typos existed. Some grammar and format mistakes should be corrected. The authors are suggested to check the manuscript as well as the supporting information.

Response: Thanks so much for the advice. We have corrected the issues as “Supplementary Figure 6. Tafel slopes for 20% Pt-C, CPF-Fe, CPF-Ni and CPF-Fe/Ni as HER catalyst in 1 M KOH”. We also have checked through the whole manuscript and the supporting information in case of such mistakes. The changes are highlighted in yellow background.

Supplementary Figure 6. Tafel slopes for 20% Pt-C, CPF-Fe, CPF-Ni and CPF-Fe/Ni as HER catalyst in 1 M KOH

Reviewer #3

This work developed the single-atom electrocatalyst with bimetallic sites, which was highly efficient for overall water splitting in wide-pH range. Impressively, this catalyst achieved demonstrate low overpotential with high stability for overall water splitting in both acidic and alkaline electrolytes. The authors have conducted substantial characterizations to verify their outcomes. The research has been well performed and the results are convincing. The pyrolysis-free synthesis for the preparation of highly efficient Fe and Ni based catalysts holds valuable novelty, which also is a good showcase for the catalytic process studies. Therefore, I would like to recommend the publication in Nature Communication. Following are some minor issues which could be addressed before final publication:

Response: Thanks so much for the comments.

1. The authors have claimed that “the pyrolysis-free route was mild and ensure the well-defined structure and precise location of active centers, offering a new path for further explanation of the mechanism”. Since such synthesis for single atomic catalysts is the important novelty of this work, the authors should provide more description or discussion for pyrolysis-free.

Response: Many thanks for the professional suggestions. In order for the deep understanding, we have added more description and discussion about pyrolysis-free in the revised manuscript. The changes have been highlighted in yellow background as followed:

Pyrolysis indeed has been very widely used in the past decades. Unfortunately, pyrolysis process are always carried out in specific atmosphere with high temperature, which are very energy-intensive. Meanwhile, the isolated metal atoms on the precursors are very unstable and tend to agglomerate during the pyrolysis process, not only reducing the efficient utilization of metal active sites but also making it difficult to realize the precise control. Thus, the development of non-carbonized synthesis strategies for the preparation of efficient electro-catalysts is challenging and vital. It could be expected that the pyrolysis-free route would lead to precisely synthetic control and a better maintained structure, holding the potentials to elucidate structure-function relationships for the guidance of optimum electro-catalysts.

2. According to the results reported by the authors, it could be concluded that the presence of Ni was very important or even essential for the catalytic performance during the overall water splitting. Thus, is it possible to supply more study about the control of Ni species during the synthesis? For example, to explore whether the Ni content can be affected by the reaction conditions.

Response: Many thanks for the professional comments. According to the questions, we have explored the effect of different reaction conditions on the Ni content. The as-synthesized CPF-Fe/Ni under different conditions were tested by ICP. According to

the results, there is no obvious affections of the different reaction conditions on the Ni content, which indicate that the contents of Ni of CPF-Fe/Ni may be saturated. The data with the conditions are listed as followed:

Table R1. ICP of CPF-Fe/Ni which was synthesized CPF-Fe/Ni under different conditions

Time (min)	Power (W)	Ratio of mass for CPF-Fe:NiCl ₂ ·6H ₂ O	Fe (%)	Ni (%)
20	400	1:1.5	11.90	1.35
20	400	1:1	11.95	1.75
20	400	2:1	12.18	1.28
20	400	1:3	11.28	1.47
40	400	1:1.5	11.68	1.69
60	400	1:1.5	12.21	1.49
20	200	1:1.5	11.90	1.37
20	800	1:1.5	11.74	1.44

3. As the authored reported, Ni single atoms were introduced by a post-synthesized. During this process, it could be noted that KOH was used in the synthesis. What the role did KOH play, and how to avoid Ni(OH)₂ was produced?

Response: Many thanks for the professional comments. According to reported work, we found that CPF-Fe support can be well dispersed under alkaline conditions. KOH could provide a good solvent for the reaction. Thus, the addition of KOH could increase the interaction between Ni ion and CPF-Fe and promote the formation of bimetallic single-atom catalysts. In the subsequent process, a certain amount of HCl was used to remove excess impurities, which further avoid the production of Ni(OH)₂. The detailed steps can be found in “Synthesis of CPF-Fe/Ni” of the manuscript. We have also added the description. The changes are marked in yellow background as followed:

It has been reported that the CPF-Fe support could be well dispersed under alkaline conditions. Thus, KOH was used as the solvent for the reaction to promote the formation of bimetallic single-atom catalysts.

4. This work involved a large number of literature comparisons, which was very necessary. These comparisons need to be guaranteed under the same conditions. The authors are suggested to clearly mark the comparison conditions.

Response: Many thanks for the professional suggestions. We have uniformed the literature comparisons to guaranteed they are all under the same conditions (10 mA

cm⁻²). We also added the detailed conditions. All the changes have been marked in Table S1 and highlighted in yellow background as followed:

Supplementary Table 1. Comparison of different electrocatalytic activity for different water splitting catalysts under 10 mA cm⁻².

	0.5 M H ₂ SO ₄ η HER/OER (mV)	1 M KOH η HER/OE R (mV)	0.5 M H ₂ SO ₄ Cell voltage (V)	1 M KOH Cell voltage (V)	Tafel (mVdec ⁻¹)	TOF (s ⁻¹)	Faradaic Efficiency (%)	Stability (h)	Ref
CPF-Fe/Ni	23/201	42/194	1.44	1.57	0.5 M H ₂ SO ₄ : 82.6 (HER) 94.1 (OER) 1 M KOH: 169.5 (HER) 102.1 (OER)	H ₂ SO ₄ : 3.16 (H ₂) 1.12 (O ₂) KOH: 2.64 (H ₂) 1.4 (O ₂)	~ 100	H ₂ SO ₄ : 120 KOH: 120	Our Work
Ru/Co-N-C	13/232	23/247	1.49	1.5	0.5 M H ₂ SO ₄ : 40.7 (HER) 1 M KOH: 32.4 (HER)	9.2	96	H ₂ SO ₄ : 20 KOH: 15	1
W-NiS _{0.5} Se _{0.5}	--	39/171	--	1.44	1 M KOH: 51 (HER) 41 (OER)	0.21	~ 100	500	2
CoNiRu-NT	--	27/255	--	1.47	1 M KOH: 78 (HER) 67 (OER)	0.330	95	48	3
FMZP4	--	53/184	--	1.79	1 M KOH: 53.2 (HER) 51.9	0.00893	~ 100	80	4

					(OER)				
Co-Co _{0.85} Se	--	97/265	--	1.47	1 M KOH: 70.7 (HER) 78.0 (OER)	4.4	~ 100	12	5
Mn ₂ P-Mn ₂ O ₃ /PNCf	--	98/330	--	1.56	1 M KOH: 46 (HER) 86 (OER)	--	~100	72	6
H-CoS _x @N iFe LDH/NF	--	95/250	--	1.98	1 M KOH: 90 (HER) 49 (OER)	0.067	97	100	7
Co-BTC	--	437/370	--	2.03	1 M KOH: 115.1 (HER) 89.1 (OER)	1.23	--	5	8
BPIr _{be} catalyst	25/290	198/290	1.57	1.54	0.5 M H ₂ SO ₄ : 30.9 (HER) 70 (OER) 1 M KOH: 91 (HER)	22 (H ₂) 4.41 (O ₂)	--	H ₂ SO ₄ : 2.5 KOH: 2.5	9
Ru/RuS ₂ -2	45/201	--	1.501	--	0.5 M H ₂ SO ₄ : 30.9 (HER) 70 (OER) 1 M KOH: 91 (HER) 64 (OER)	0.71 (H ₂) 0.61 (O ₂)	--	10	10
Ru1/D-NiFe LDH	--	10/189	--	1.419	1 M KOH: 29 (HER) 31 (OER)	7.66 (H ₂)	~100 (H ₂) 99.6 (O ₂)	2.5	11

D-CoP-HoMSs	--	93/294	--	1.57	1 M KOH: 50 (HER) 67 (OER)	--	--	-	12
NiCoPO@NC/P-NF-e	--	73.1/221.4	--	1.5	1 M KOH: 82 (HER) 87.8 (OER)	0.21 (O ₂)	98.5 (H ₂) 99.4 (O ₂)	48	13
Ni _{0.6} Fe _{0.4} -MOG	--	159/285	--	1.61	1 M KOH: 38 (HER) 63 (OER)	1.44(H ₂) 1.38(O ₂)	~ 100	20	14
Fe doped MOF CoV@CoO nanoflakes	--	78/220	--	1.53	1 M KOH: 86 (HER) 59 (OER)	0.45	--	50	15
NiMoOx/NiMoS	--	38/186	--	1.46	1 M KOH: 38 (HER) 34 (OER)	1.97	99.6±0.3 (H ₂) 97.5±0.4 (O ₂)	500	16
CoFeO@BP	--	88/256	--	--	1 M KOH: 51 (HER) 42 (OER)	--	~ 100 (H ₂) 99.2 (O ₂)	24	17
CoP-InNC@CNT	--	159/270	--	1.58	1 M KOH: 63 (HER) 85 (OER)	--	--	15	18
Ni@N-HCG HF	--	95/260	--	1.6	1 M KOH: 57 (HER) 63 (OER)	--	--	20	19
Co ₂ P/CoNPC	--	208/328	--	1.64	1 M KOH: 83.0 (HER) 72.6 (OER)	--	--	8.3	20
D-Ni-MOF NSA	--	101/219	--	1.5	1 M KOH: 50.9	1.224	--	48	21

					(HER) 48.2 (OER)				
FeNi(BDC) (DMF,F)/N F	--	234/227	--	1.58	1 M KOH: 96.2 (HER) 37.4 (OER)	0.298	~ 100	30	22
HOF-Co0.5 Fe0.5/NF	--	170/278	--	1.63	1 M KOH: 137 (HER) 59 (OER)	--	99.9	20	23

5. Given to the scientificity and the continuity of the data, the video of overall water splitting process is more intuitive and easy to accept. The authors are suggested to provide the videos of the water splitting.

Response: Many thanks for the professional suggestions. The videos of overall water splitting process in 0.5 M H₂SO₄ and 1M KOH have been provided in the revised version as supplied information and referred in the related part of the manuscript. The changes are all marked in yellow background.

...which was performed in a sealed H-type cell wherein CPF-Fe/Ni was acted as both cathodic and anodic electrodes (Supporting Information Fig.23 and 25, Video 1 and 2).

In the revised supporting information:

Caption of video 1. The video of overall water splitting process in 0.5 M H₂SO₄.

Caption of video 2. The video of overall water splitting process in 1M KOH.

6. Some typos and errors in grammar and format need to be corrected: On Page 3, Line 9, “have attracted numerous attentions” should be corrected for “have attracted numerous attention”. On Page 4, Line 11, “various routes have been developed to improve the atomic utilization over the past decades.” should be corrected for “various routes had been developed to improve the atomic utilization over the past decades”. Similar mistake also found on Page 5, Line 7 and 8. The format of references should be unified. Such as in Ref 32 “Energy Environ. Sci., 15” should be “Energy Environ. Sci. 15”. Meanwhile, in Ref 10 “CO2” should be “CO₂”.

Response: Many thanks for the careful suggestions. We have corrected the mentioned typos, grammar issues and formats of references. We also have checked through the whole manuscript and the supporting information in case of such mistakes. The changes are highlighted in yellow background.

“have attracted numerous attentions” have been corrected for “have attracted numerous attention”

“various routes have been developed to improve the atomic utilization over the past decades.” have been corrected for “various routes had been developed to improve the atomic utilization over the past decades”.

“Energy Environ. Sci., 15” has been corrected for “Energy Environ. Sci. 15”

“CO₂” has been corrected for “CO₂”

REVIEWERS' COMMENTS

Reviewer #1 (Remarks to the Author):

The revised manuscript can be accepted now.

Reviewer #2 (Remarks to the Author):

After the revision, this work could be accepted.

Reviewer #3 (Remarks to the Author):

Comments

This work developed the single-atom electrocatalyst with bimetallic sites, which was highly efficient for overall water splitting in wide-pH range. Impressively, this catalyst achieved demonstrate low overpotential with high stability for overall water splitting in both acidic and alkaline electrolytes. The authors have conducted substantial characterizations to verify their outcomes. The research has been well performed and the results are convincing. The pyrolysis-free synthesis for the preparation of highly efficient Fe and Ni based catalysts holds valuable novelty, which also is a good showcase for the catalytic process studies. All comments by the reviewers have been adressed in detail, thus the manuscript can be accepted for publication.

Point by Point Response

Reviewer #1

The revised manuscript can be accepted now.

Response: Thanks so much for the reviewers' efforts and a professional reviewing.

Reviewer #2

After the revision, this work could be accepted.

Response: Thanks so much for the reviewers' efforts and a positive reviewing.

Reviewer #3

This work developed the single-atom electrocatalyst with bimetallic sites, which was highly efficient for overall water splitting in wide-pH range. Impressively, this catalyst achieved demonstrate low overpotential with high stability for overall water splitting in both acidic and alkaline electrolytes. The authors have conducted substantial characterizations to verify their outcomes. The research has been well performed and the results are convincing. The pyrolysis-free synthesis for the preparation of highly efficient Fe and Ni based catalysts holds valuable novelty, which also is a good showcase for the catalytic process studies. All comments by the reviewers have been adressed in detail, thus the manuscript can be accepted for publication.

Response: Thanks so much for the reviewers' attention and a professional reviewing.